# Primary Effusion Lymphoma: A Rare and Challenging Diagnosis for Recurrent Pleural Effusion

**DOI:** 10.3390/diagnostics13030370

**Published:** 2023-01-19

**Authors:** Letícia Jacome Pereira, Sara Mohrbacher, Precil Diego Miranda de Menezes Neves, Flavia Fernandes Silva Zacchi, Ivan Ucella Dantas Medeiros, Victor Augusto Hamamoto Sato, Érico Souza Oliveira, Leonardo Victor Barbosa Pereira, Américo Lourenço Cuvello-Neto, Otávio Baiocchi, Pedro Renato Chocair

**Affiliations:** 1Internal Medicine Service, Oswaldo Cruz German Hospital, São Paulo 01323-020, Brazil; 2Pathology Service, Fleury Laboratory, São Paulo 01431-001, Brazil; 3Oncology Center, Oswaldo Cruz German Hospital, São Paulo 01323-020, Brazil

**Keywords:** cancer, hematology, lymphoma, oncology, pathology, pleural effusion

## Abstract

Primary Effusion Lymphoma is an extremely rare and aggressive subtype of B-cell lymphoma, accounting for only <1% of all cases of this neoplasm. It has a unique clinical presentation because it has a predilection for appearing in body cavities, such as the pleural space, pericardium and peritoneum. It mainly affects immunocompromised individuals and may also affect individuals in the Mediterranean region and in areas endemic for human herpesvirus 8 (HHV-8). Herein, we report the case of an 83-year-old immunocompetent male complaining of coughing, fever and progressive dyspnea for 3 days. His past medical history revealed a recurrent pleural effusion for the last three years, as well as losing weight and malaise. A subsequent investigation revealed a PEL diagnosis of the pleura.

**Figure 1 diagnostics-13-00370-f001:**
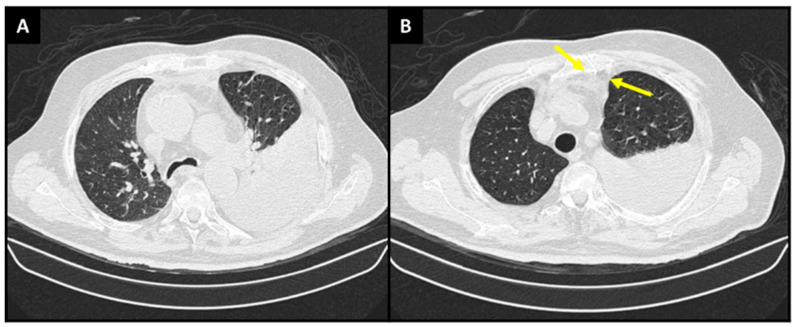
Chest Computed Tomography scan showing (**A**) a large left pleural effusion and (**B**) densification of the anterior mediastinal fat (yellow arrows), a finding that suggests pleural inflammation. Pleural effusion (PE) is a frequent entity in clinical practice, with an estimated prevalence of 400 cases per 100,000 inhabitants and a global incidence of 1.5 million cases per year [1,2]. Many diseases may manifest themselves as pleural effusion, including hypervolemia, infections, autoimmune diseases, bleeding, drug use and neoplasms. In this scenario, the analysis of clinical history, comorbidities, complementary exams and careful evaluation of the pleural fluid is of extreme importance to diagnose and treat this condition associated with cavity effusion [3,4,5]. The most common causes of pleural effusion are congestive heart failure, infectious diseases and cancer [3,4]. Primary Effusion Lymphoma (PEL) is an extremely rare and aggressive subtype of B-cell lymphoma, accounting for only <1% of all cases of this neoplasm. PEL has a unique clinical presentation because it has a predilection for appearing in body cavities, such as the pleural space, pericardium and peritoneum. It mainly affects immunocompromised individuals and may also affect individuals in the Mediterranean region and in areas endemic for human herpesvirus 8 (HHV-8). It is diagnosed by exclusion, mainly being given by an analysis of the cavity fluid [6,7,8]. Herein, we report the case of an immunocompetent patient with recurrent pleural effusion for three years, with an investigation revealing the diagnosis of pleural PEL. An 83-year-old male patient born in Italy and coming from Brazil attended an emergency service complaining of coughing, fever and progressive dyspnea for 3 days. He denied similar symptoms in family members or exposure to people with a confirmed diagnosis of COVID-19. He reported that he had had several episodes of hospitalization for a similar condition in the last three years, with pleural effusion always being detected and treated with diuretics; however, no further investigation was performed. He had previous diagnoses of type 2 diabetes mellitus, hypertension, was a former smoker (60 years/one pack a day) and was on regular use of Metformin, Gliclazide, Dapaglifozin, Furosemide, Enalapril, Amlodipine and Acetylsalicylic Acid. He was in regular general condition according to the physical examination upon admission, emaciated, dyspneic (RR: 24 ipm) and pulmonary auscultation was abolished up to the medical third on the left, suggesting pleural effusion. SO_2_ was 88% in room environment air. There was no lower limb edema or jugular swelling. The rest of the physical examination was unremarkable. Laboratory tests showed the following results: Hemoglobin: 11.1 g/dL; Hematocrit: 35.9%; Leukocytes: 8870/mm^3^; Eosinophils: 7010/mm^3^; Platelets: 340,000/mm^3^; Urea: 26 mg/mL; Creatinine: 0.64 mg/dl; no hydroelectrolytic or acid-base disorders; C-Reactive Protein: 6 mg/dL; D-dimer: 869 ng/mL; LDH: 320 U/L; BNP: 33 pg/mL; PCR for COVID-19: Negative; and Pro-calcitonin: <0.12. In addition, hepatitis B, C and HIV serology were negative. Echocardiogram showed an ejection fraction of 67%, without signs of pericardial effusion. Chest tomography showed bilateral pleural effusion, being discreet on the right and voluminous on the left, determining atelectasis of almost the entire left lower lobe and lingula associated with densification of the anterior mediastinal fat, with intermingled liquid layers (Figure 1A,B). The patient underwent a left thoracentesis with drainage of 1700 mL of liquid with a yellow-citrus appearance. Biochemical analysis of the pleural fluid revealed an exudate pattern, with pH of 7.7 and ADA of 169 U/L, with negative PCR for Mycobacterium tuberculosis, as well as cultures for aerobic and anaerobic bacteria, fungi and mycobacteria. The cytological evaluation detected 1672 cells/mm^3^, with a predominance of lymphocytes (48%) and macrophages (28%). A description of atypical mononuclear cells with a monotonous pattern, along with the presence of intense cellular atypia, evident nucleoli and numerous mitotic figures, was also evidenced by cell block analysis (Figure 2A,B). The pleural fluid was then submitted to immunohistochemistry for further investigation, with the following findings: lymphoid population with immunoblastic/plasmablastic morphology; co-expression of EMA, MUM1 and HHV-8 (Figure 2C), in addition to weak and partial expression of CD45, CD43 and CD30; Ki 67: 70%; and negative stain for EBER. A PET-CT was performed, which did not detect anomalous glucose uptake in whole-body analysis. The association of clinical findings and complementary exams was compatible with Primary Pleural Effusion Lymphoma.

**Figure 2 diagnostics-13-00370-f002:**
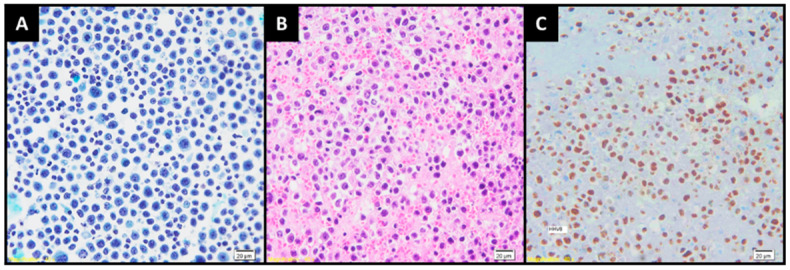
Cytological analysis of pleural fluid showing: (**A**) mononuclear cells with a monotonous appearance, with the presence of atypia, evident nucleolus and mitosis figures; alterations which were also evidenced in (**B**) cell-block material; (**C**) Immunohistochemical analysis revealed nuclear positivity for anti-HHV8 antibody, a defining point of the diagnosis. In view of the patient’s age, comorbidities and performance status, it was decided to initiate an R-miniCHOP regimen (Rituximab + Cyclophosphamide + Doxorubicin + Vincristine + Prednisolone). There was a relapse of the disease after the fourth chemotherapy cycle, with an association of pleural effusion and massive ascites, and so it was decided to switch to a COP regimen (Cyclophosphamide, Vincristine and Prednisone). There was no response with the second regimen, so we opted for a third line of treatment with metronomic chemotherapy with cyclophosphamide + etoposide + oral dexamethasone, associated with paracentesis and relief thoracentesis. The patient evolved with severe dysphagia and probable aspirative pneumonia; thus, end-of-life care was chosen in agreement with the family. The patient then evolved to death from septic shock of pulmonary focus. Although pleural effusion is a relatively common finding in patients with lymphoma, PEL is a unique entity, as it is primarily an effused tumor in the body cavities such as the pleural, pericardial and peritoneal spaces. In most cases, it usually occurs in the absence of identifiable tumor masses or lymphadenopathy [7,8,9]. According to the current World Health Organization (WHO) classification [10,11], PEL is a rare distinct disease entity of large B-cell lymphoma that affects preferentially immunocompromised men, most often by the human immunodeficiency virus (HIV) [12,13,14,15], and may also affect individuals with reduced immunity due to other causes such as cirrhosis, hemodialysis or transplant patients [16,17,18,19]. In rare cases, PEL can affect older adults who are immunocompetent but live in geographic areas with a high prevalence of HHV-8, such as the Mediterranean region [20,21,22,23]. The clinical picture is characteristic of effusions, changing according to the affected area. Patients, such as the one described above, with pleural involvement may present dyspnea, chest pain, coughing or fever. Although rare, pleural effusion lymphoma can also present the presence of solid tumors. These extracavitary presentations most commonly involve the gastrointestinal tract. The causes of death are associated with disease progression or the evolution of the underlying disease in the case of immunocompromised patients [9,14,21,23,24,25]. By definition, PEL cases must have evidence of Kaposi’s sarcoma-associated herpesvirus infection, also known as HHV-8. HHV-8 is a member of the gamma herpes virus family, which includes EBV. HHV-8 is a linear double-stranded DNA virus that is not ubiquitous, but has endemic areas of infection, including sub-Saharan Africa and the Mediterranean region, while North America only exhibits an infection rate of 1% to 3% among asymptomatic blood donors. The HHV-8 virus infects host B cells, interfering with signaling pathways, cell death, inflammatory processes and immune response [24,26,27]. The exact mechanism by which HHV-8 promotes oncogenesis in PEL is an area of active investigation. HHV-8 genomes exist in PEL cells as monoclonal or oligoclonal episomes. Most infected cells express a latent pattern of gene expression, while a very small percentage express genes characteristic of the lytic phase. Infected cells can undergo clonal expansion, even with the expression of latent genes, eventually leading to neoplastic transformation by mechanisms of increased proliferation and decreased apoptosis. While most PEL cases show evidence of infection with EBV in addition to HHV-8, EBV plays an unclear role in PEL oncogenesis [6,13,28,29]. A pleural effusion lymphoma diagnosis Is performed by exclusion. The first hypothesis to be ruled out is always cavity effusion due to neoplasia of other sites, and therefore, imaging tests such as chest and abdomen tomography, PT-scan and laboratory tests such as hemogram should be requested in addition to a cavity fluid analysis [9,24,30]. The diagnosis of PEL is based on the pathological analysis of the involved tissue. Neoplastic cells are morphologically large, have rounded to irregular nuclei, prominent nucleoli and variable amounts of cytoplasm, which is occasionally vacuolated. Cells exhibit a variety of appearances from immunoblastic to plasmablastic to anaplastic. Anaplastic forms include multinucleated and Reed–Sternberg-like cells. Detecting evidence of HHV-8 viral infection in neoplastic cells is essential for diagnosing PEL. Although serology is the best way to assess whether a patient has been previously infected with HHV-8, immunohistochemical staining for LANA-1 is the standard assay for detecting evidence of HHV-8 infection in tissue; immunophenotypically, PEL cells usually exhibit a “null” lymphocyte phenotype, meaning that CD45 is expressed, but routine B cell markers (including surface and cytoplasmic immunoglobulin, CD19, CD20, CD79a) and T cells (CD3, CD4, CD8) are absent. Instead, several lymphocyte activation markers (CD30, CD38, CD71, human leukocyte antigen DR) and plasma cell differentiation (CD138) are usually displayed. Finally, cytogenetic analysis reveals complex karyotypes but no chromosomal abnormalities common in PEL [10,20,26,28,29,30,31]. Pleural effusion lymphoma has no standard treatment, and its prognosis is poor; standard chemotherapy is used in most cases, and the median survival is 6 months when there is no significant response. In cases of patients with HIV infection, it is always necessary to start treatment for the underlying disease if it has not been started. Some cases have demonstrated prolonged remissions with bortezomib-containing regimens [6,13,24]. Another successful chemotherapy regimen for the treatment of PEL is the dose-adjusted EPOCH (cyclophosphamide, doxorubicin, etoposide, vincristine, prednisone). This scheme has also proven to be effective to treat PEL, even in HIV patients [30,32,33,34]. In conclusion, Primary Effusion Lymphoma is a rare neoplasm, and it is even more difficult to be found in immunocompetent patients by laboratory diagnosis and still without specific treatment. The disease should be remembered as a differential diagnosis of recurrent pleural effusion.

## Data Availability

All of the patient’s data are presented in this report.

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
