# Peer review of "Primary Effusion Lymphoma: A Rare and Challenging Diagnosis for Recurrent Pleural Effusion"

_diagnostics, 2023, doi:10.3390/diagnostics13030370_

Round 1

Reviewer 1 Report

Pereira et al. reported a case of PEL arising in 83-year-old immunocompetent male.

This reviewer could not understand the novelty of this case. Please clearly describe the point which the authors would like to emphasize. 

Author Response

RESPONSE TO REVIEWER 1 COMMENTS

Dear reviewer,

Thank you so much for your revision and relevant insights. Please find following the answers to your raised points:

Point 1: This reviewer could not understand the novelty of this case. Please clearly describe the point which the authors would like to emphasize.

Response 1: Primary Efusion Lymphoma is a rare entity that predominantly affects immunosuppressed patients. So, describing a case in an immunocompetent patient confirms its importance. In our case, the diagnostic workup was performed by a clinician, on a patient admitted to the emergency room several times previously and without a diagnosis of the etiology of the pleural effusion. Besides, the patient was referred for specialized treatment with a hematologist for treatment only after the diagnostic was made. This case serves as a warning to clinicians for the differential diagnose of less frequent causes of pleural effusion in the emergency room.

Point 2: Moderate English changes required 

Response 2: The article was revised by a proofreading service.

Sincerely;

Dr. Precil Menezes

Reviewer 2 Report

This is an interesting case report of primary effusion lymphoma in an immunocompetent patient and its diagnosis and treatment. I would recommend the publication of this manuscript.

There are some minor suggestions to the authors.

1) line 56, “DHL” is probably LDH.

2) Was EBER performed in the specimen?

3) The authors can briefly discuss the treatment of primary effusion lymphoma such as dose-adjusted (DA) EPOCH (etoposide, prednisone, vincristine, cyclophosphamide, and doxorubicin) in the discussion section.

Author Response

RESPONSE TO REVIEWER 2 COMMENTS

Dear reviewer,

Thank you so much for your revision and relevant insights. Please find following the answers to your raised points:

Point 1: line 56, “DHL” is probably LDH.

Response 1: Thank you for your raised point. We made the correction as suggested.

Point 2: Was EBER performed in the specimen?

Response 2: Thank you for your question. EBER staining was performed by immunohistochemistry as was negative. We added this information on the revised form of the manuscript.

Point 3: The authors can briefly discuss the treatment of primary effusion lymphoma such as dose-adjusted (DA) EPOCH (etoposide, prednisone, vincristine, cyclophosphamide, and doxorubicin) in the discussion section.

Response 3: Thank you for your point. We briefly discussed about DA-EPOCH in the discussion section with its respective reference.

Point 4: English language and style are fine/minor spell check required

Response 4: The article was revised by a proofreading service.

Sincerely;

Dr. Precil Menezes

Round 2

Reviewer 1 Report

Thank you very much for your reply.